# AUDOFORMER: AN EFFICIENT TRANSFORMER WITH CONSISTENT AUXILIARY DOMAIN FOR SOURCE-FREE DOMAIN ADAPTATION

## ABSTRACT

Source-free domain adaptation (SFDA), which tackles domain adaptation without accessing the source domain directly, has gradually gained widespread attention. However, due to the inaccessibility of source domain data, deterministic invariable features cannot be obtained. Current advanced methods mainly evaluate pseudo-labels or consistent neighbor labels for self-supervision, which are susceptible to hard samples and affected by domain bias. In this paper, we propose an efficient trans**Former** with a consistent **Au**xiliary **do**main for source-free domain adaptation, abbreviated as **AudoFormer**, which solves the invariable feature representation from a new perspective by domain consistency. Concretely, Audo-Former constructs an auxiliary domain module (ADM) block, which can achieve diversified representations from the global attention feature in the intermediate layers. Then based on the auxiliary domain and target domain, we distinguish invariable feature representation by exploiting multiple consistency strategies, i.e., dynamically evaluated consistent labels and consistent neighbors, which can divide the whole target samples into source-like easy samples and target-specific hard samples. Finally, we align the source-like with the target-specific samples by conditional guided multi-kernel max mean discrepancy (CMK-MMD), which guides the hard samples to align the corresponding easy samples. To verify the effectiveness, we conduct extensive experiments on three benchmark datasets (i.e., Office-31, Office-Home, and VISDA-C). Results show that our approach achieves significant performance among multiple domain adaptation benchmarks compared to the other state-of-the-art baselines. *Code will be available.*

## 1 INTRODUCTION

Deep learning has achieved remarkable success in various applications of computer vision but also suffers from some drawbacks. Firstly, the success of deep learning relies on huge manual annotation, which is extremely expensive in real-world applications. Besides, owing to insufficient generalization, applying existing models to other relevant scenarios usually encounters the domain shifting. To address the said issue, unsupervised domain adaptation (UDA) has been introduced, which aims to transfer knowledge from the source domain to the target domain. The key idea of UDA is to exploit the network to extract the labeled source domain and unlabeled target domain feature, and then project the feature in common feature space to learn domain invariant Long et al. (2015) and common semantic information Goodfellow et al. (2014). Nevertheless, since security privacy protection and data transmission limitations, the model cannot access the source data directly in some scenarios.

To overcome the data-absent problem, some prospective efforts Liang et al. (2020); Yang et al. (2021b); Saito et al. (2018); Li et al. (2020); Xia et al. (2021) work on source-free domain adaptation (SFDA). In general, these methods roughly fall into two branches: adversarial-based approaches and pseudo-label evaluation. Adversarial-based approaches mainly utilize the generator to synthesize the target feature distribution and aim to improve the generalization performance of the target domain Saito et al. (2018); Li et al. (2020); Xia et al. (2021). However, these approaches incur extensive resources and time to optimize the generator and synthetic images, and the generated samples cannot fully reflect the distribution of the real data when encountering complex data sce-

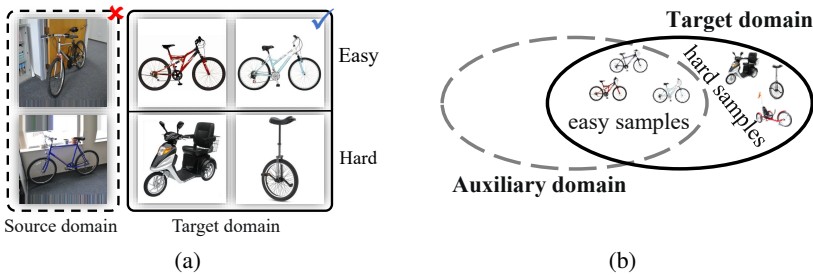

Figure 1: Basic thinking of our proposed methods. (a): The above are easy samples, which are more similar to the source data. The below are hard samples, which are hard to discriminate. (b): Construct an auxiliary domain to discriminate easy and hard samples.

narios, thereby limiting the generalization performance. While pseudo-label evaluation approaches commonly leverage the pre-trained model of the source domain to evaluate the centroid by clustering the nearest neighborhood feature, and mitigate variation between cross-domains by certain constraints Liang et al. (2020); Yang et al. (2021b). Although this method has achieved promising results, it is still vulnerable to erroneous evaluations caused by hard samples. For example, in the Office31 dataset as Fig. 1 (a), easy samples are closer to the source domain while hard samples have great discrepancies. These hard samples are difficult to align and are more prone to produce incorrect evaluations. Compared to DA tasks Long et al. (2015), the main dilemma of SFDA is the inability to obtain invariant feature representations. Intuitively, if obtaining sufficient relevant features, we can emulate the invariant features to reconstruct the source domain with some strategies, as shown in Fig. 1 (b). Therefore, this motivates us to pay attention to the intermediate layer features.

In this paper, we proposed an efficient trans**Former** with a consistent **Au**xiliary **do**main for source-free domain adaptation (termed as **AudoFormer**), which solves the invariable feature representation from a new perspective by domain consistency. We exploit the vision transformer (ViT) as our backbone model mainly for the following considerations: First, compared to the CNN-based model, the inner feature dimensions of each layer in the ViT backbone are consistent, enabling the relatively low cost of constructing the auxiliary domain. Second, we argue that self-attention features can capture discriminative features, which surpass general semantic features in terms of representation power. Besides, since the long-distance dependence of features is solved, the current ViT model has achieved immense success in many fields Dosovitskiy et al. (2021); Carion et al. (2020); Zheng et al. (2021), while related research has been relatively less explored in the field of SFDA. This motivates us to adopt the ViT backbone model and its variants to study related problems. Specifically, AudoFormer first acquires the multi-layer global attention from the intermediate layers via exponential moving average (EMA), and then introduces an auxiliary domain module (ADM) block for the ViT backbone to obtain additional diverse representations from the global attention, which can be used to construct an auxiliary domain, as shown in Fig. 2. We train the AudoFormer in the source stage (i.e., first stage) by supervised learning and self-distillation. Then we initialize the target model with the parameters of the source domain, fixing the auxiliary domain module and the target domain module to conduct domain adaptation. In light of the auxiliary domain and target domain of domain adaptation stage (i.e., second stage), we distinguish invariable feature representation by employing multiple consistency strategies, i.e., dynamically evaluated consistent labels and consistent neighbors, which can divide the whole target samples into source-like easy samples and target-specific hard samples. Finally, we align the source-like with the target-typical sample by conditional guided multi-kernel max mean discrepancy (CMK-MMD), which guides the hard samples to align corresponding easy samples.

To verify the effectiveness, we conduct extensive experiments on three benchmark datasets, i.e., Office-31, Office-Home, and VISDA-C. Eventually, our approach can outperform other source-free approaches and achieve superior performance among multiple domain adaptation benchmarks. To sum up, our key contributions are as follows: 1) We introduce an efficient transformer framework with an auxiliary domain for SFDA and first solve the issue from a new perspective by domain consistency. 2) We propose to exploit multiple consistency strategies to distinguish invariable feature representation which can divide the whole target samples into score-like easy samples and target-specific hard samples. 3) To further improve the alignment effect of the domain adaptation, we align the source-like with target-specific samples by CMK-MMD, which guides the samples to align the

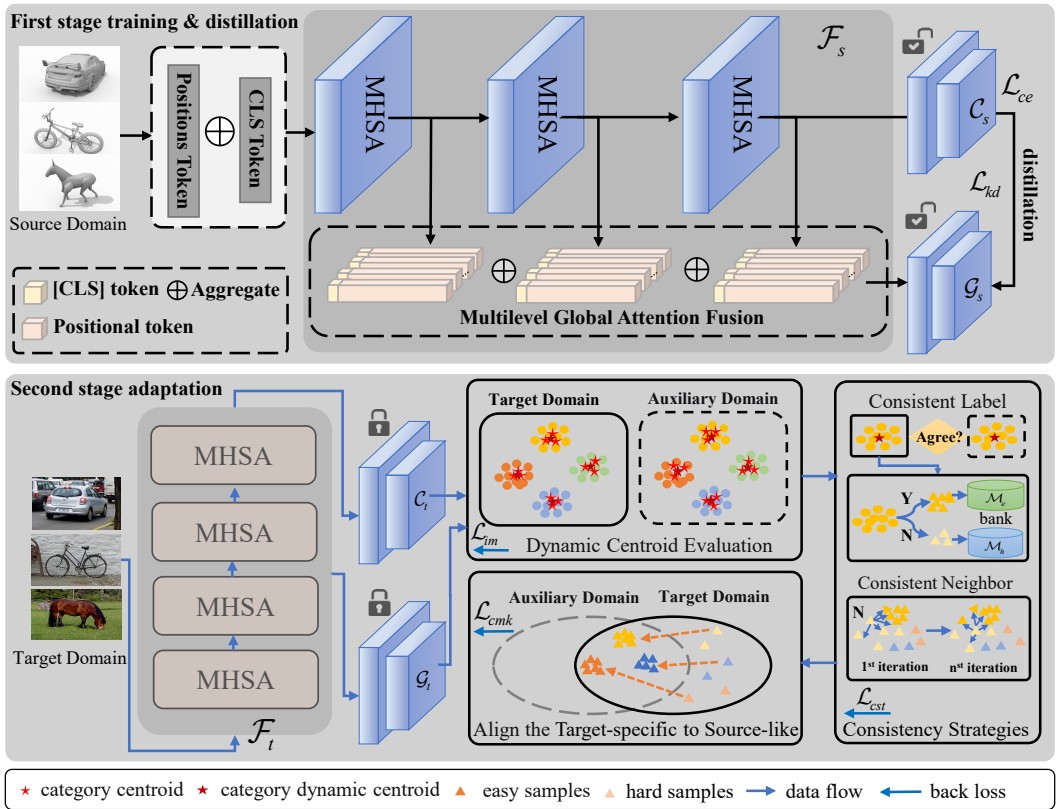

Figure 2: The overall workflow of the proposed AudoFormer. The **Upper** is the first-stage training and distillation in the source domain, where the attention feature of each layer is aggregated by EMA to produce global attention. The ADM block $\mathcal{G}_s$ is the trainable module by distillation of the output of classifier $\mathcal{C}_s$. The **Bottom** is the adaptation stage. We first evaluate the pseudo-labels of the target domain and auxiliary domain to distinguish easy samples (dark color) and hard samples (light color) by consistency strategies. Then, we store the easy and hard samples in the memory bank respectively and re-evaluate the hard samples by consistent neighbors. Finally, we exploit CMK-MMD to align the samples of source-like and target-specific.

hard samples to corresponding easy samples. 4) Extensive experiments are conducted on three benchmark datasets to verify the effectiveness of our proposed approaches.

## 2 METHODS

### 2.1 PRELIMINARIES AND NOTATIONS

**Notation.** In this paper, we address the unsupervised SFDA task only with a pre-trained source model and without accessing any source data. We consider image classification and denote the labeled source domain data with $n_s$ samples as $\mathcal{D}_s = \left\{ (x_s^i, y_s^i) \right\}_{i=1}^{n_s}$ where $x_s^i \in \mathcal{X}_s, y_s^i \in \mathcal{Y}_s$, and unlabeled target domain data with $n_t$ samples as $\mathcal{D}_t = \left\{ x_t^i \right\}_{i=1}^{n_t}$ where $x_t^i \in \mathcal{X}_t$. The goal of SFDA is to train a classifier of $\mathcal{F} \circ \mathcal{C}$ with the given source data to predict the labels $\left\{ y_t^i \right\}_{i=1}^{n_t}$ of unlabeled data in the target domain, where $\mathcal{F}(\cdot; \theta_{\mathcal{F}}) : x_t^i \to f_t^i$ denotes the backbone model of features extractor, where $f_t$ denotes the feature representation. $\mathcal{C}(\cdot; \theta_{\mathcal{C}}) : f_t^i \to z_t^i$ denotes the linear classifier, and $z_t^i$ represents the distribution of logits, as shown in Fig. 2. Besides, we introduce an ADM block as $\mathcal{G}(\cdot; \theta_{\mathcal{G}}) : \hat{x}_a^i \to \hat{f}_a^i$ and $\hat{z}_a^i$, where $\hat{x}_a^i$ denotes global attention feature, $\hat{f}_a^i$ and $\hat{z}_a^i$ denotes the feature representations and the distribution logits extracted by the ADM, respectively.

**Multi-head Self-Attention.** Self-attention mechanism is a core component of ViT, which captures the long-distance dependence by MHSA. For self-attention, the layer-normalized patches

are mapped by three learnable linear projectors into vectors, i.e. queries $\mathbf{Q} \in \mathcal{R}^{(N+1)\times D}$, keys $\mathbf{K} \in \mathcal{R}^{(N+1)\times D}$ and values $\mathbf{V} \in \mathcal{R}^{(N+1)\times D}$. N donates the length of patch sequence, D indicates the dimensions of Q and K. Then the queries and keys are matched by the inner product to have an $(N + 1) \times (N + 1)$ matrix, which denotes the semantic relevance of the query-key pairs in the corresponding position and applies a softmax function to obtain the weights on the values, which is given by:

$$\mathrm{SA}(\mathbf{Q}, \mathbf{K}, \mathbf{V}) = softmax(\frac{\mathbf{Q}\mathbf{K}^{\mathrm{T}}}{\sqrt{D}}) \cdot \mathbf{V} \,, \tag{1}$$

$$\mathrm{MHSA}(\mathbf{Q}, \mathbf{K}, \mathbf{V}) = \mathrm{Concat}(\mathrm{SA}_1, ..., \mathrm{SA}_k) \,,$$

Concat here represents a concatenation operation. In our solution, we learn the representations from the multilevel global attention fusion as detailed in the following.

## 2.2 AUXILIARY DOMAIN MODULE BLOCK

Owing to the inability to access the source domain data, compared to DA tasks Long et al. (2015), the main dilemma of SFDA is the inability to obtain invariant feature representations. Therefore, if obtaining diverse representations from the existing source domain, we can distinguish invariant feature representations by some strategies, which can be employed to discriminate source-like easy samples and target-specific hard samples. To obtain the diverse representations, we pay attention to the intermediate layers, which are widely exploited in current research Dosovitskiy et al. (2021); Chen et al. (2023); Liang et al. (2022). We exploit the ViT models as our backbone. Compared with the CNN models, ViT models excel in addressing the long-distance dependence issue of features, however, missing a certain inductive bias makes them require a large dataset (e.g., ImageNet ILSVRC 2012 dataset Deng et al. (2009) or JFT-300M dataset Sun et al. (2017)) to compensate for this. Therefore, to address the shortcoming of inductive bias and achieve diverse representations, we introduce an extra CNN-based auxiliary domain module for AudoFormer, which allows us to exploit the feature representations from the intermediate MHSA layers Dosovitskiy et al. (2021); Chen et al. (2023); Liang et al. (2022). ViT model exploits the class token (i.e., [CLS] token) to learn the global positional information of all patches and utilize it for the final classification of objects. However, some research Liang et al. (2022); Chen et al. (2023) have proven that not all layers of attention can focus on discriminative objects. If directly achieving the feature of each layer will lead to a weak representation. Therefore, we take the features of the multilevel MHSA into account and aim to learn the feature representations from global self-attention. Specifically, we aggregate the attentional features of each layer by exponential moving average (EMA) to achieve the global attention as $\hat{x}_a = \lambda \cdot \hat{f}_l^a + (1 - \lambda) \cdot \hat{f}_{l-1}^a$, where $\lambda = 0.99$ and $l$ denotes the current layer of MHSA. We follow the suggestion of Liang et al. (2022) and set $l > 4$-th layer to ensure the acquisition of effective attention features. Then, our AudoFormer extracts the feature of the global attention by the auxiliary module as

$$\hat{e}_1 = \mathrm{ReLU}(\mathrm{BN}(\mathrm{Cov}^{3\times 3}(\hat{x}_a))) \tag{2}$$

$$\hat{e}_2 = \mathrm{ReLU}(\mathrm{BN}(\mathrm{Cov}^{3\times 3}(\hat{e}_1)))$$

$$\hat{f}_a = \mathrm{AvP}(\mathrm{BN}(\mathrm{Cov}^{3\times 3}(\hat{e}_2)))$$

$$\hat{z}_a = Linear(ReLU(Linear(\hat{f}_a)))$$

where $\hat{e}_1$ and $\hat{e}_2$ indicate the internal latent vectors of the ADM, $\mathrm{Cov}^{3\times 3}$ denotes the convolution with a kernel size of $3 \times 3$, BN, ReLU, and AvP denote the batchnormal layer, rectified linear unit, and average pool layer, respectively. The ADM block is trained in the first stage of the source domain and fixed for adaptation of the target domain, which can extract the feature representations $\hat{f}_a$ and distribution logits $\hat{z}_a$ to construct the auxiliary domain.

### 2.2.1 FIRST STAGE TRAINING AND SELF-DISTILLATION

In the first stage of our framework, we train the model in the source domain with ground truth by standard cross-entropy loss and aim to obtain the optimal ADM parameters $\theta_{\mathcal{G}}^*$ by minimizing the discrepancy between ADM $\mathcal{G}$ and source classifier $\mathcal{C}$ as

$$\theta_{\mathcal{G}}^* = \underset{\theta_{\mathcal{G}}}{\arg\min} \, \mathbb{E}_x[\mathcal{D}(\mathcal{G}(\hat{x}_a; \theta_{\mathcal{G}}), \mathcal{F}(x_s; \theta_{\mathcal{F}}) \circ \mathcal{C}(f_s; \theta_{\mathcal{C}}))] \,, \tag{3}$$

where the $\theta_{\mathcal{G}}^*$ denotes the optimized weight of the ADM block, which is trainable in the source domain. To realize this, we exploit the self-distillation strategy to optimize the weight $\theta_{\mathcal{G}}^*$ of ADM,

which distills the knowledge from the feature $f_s^i$ and logits $z_s^i$ of classifier $\mathcal{C}$. Therefore, in the source domain, we train the model by the formula as follows:

$$\mathcal{L}_{kd}^s = \frac{1}{n_s} \sum_{i=1}^{n_s} \left( \|\hat{f}_{a,s}^i - f_s^i\|_2 + \hat{z}_{a,s}^i \cdot \mathbf{log} \frac{\hat{z}_{a,s}^i}{z_s^i} \right) , \tag{4}$$

where $\hat{f}_{a,s}^i$ and $\hat{z}_{a,s}^i$ denote the feature and logits mapped by ADM in the source domain. According to our experiments in Tab 4, AudoFormer can significantly improve the performance of the source domain. This demonstrates the ability of AudoFormer to mitigate the inductive bias. We then apply it to the following domain adaptation in the second stage.

## 2.3 MULTIPLE CONSISTENCY STRATEGIES

In the domain adaptation progress, we first exploit AudoFormer to extract the feature representations $f_t$ and $\hat{f}_{a,t}$ with corresponding logits distribution $z_t$ and $\hat{z}_{a,t}$ from unlabeled dataset $\mathcal{X}_t$. Then, we freeze the source classifier and ADM, i.e., $\mathcal{C}$ and $\mathcal{G}$. If the pseudo-labels of some samples are calculated directly, the noise may affect them. Therefore, the consistency strategies are exploited to discriminate invariable feature representation, which can divide the samples into easy and hard samples and ensure the pseudo-labels of samples with higher confidence.

### 2.3.1 DYNAMIC CENTROID EVALUATION

The centroid evaluation proposed by Liang et al. (2020) has proven to be effective in practice. However, if exploring the centroid in each iteration, we found that it was easily disturbed by noise. Thus, differing from the traditional approaches Liang et al. (2020; 2021), we exploit the dynamic strategy to calculate the initial centroid of each category by

$$c_k^j = \frac{\sum_{x_t \in \mathcal{X}_t} exp(z_t^j) f_t^j}{\sum_{x_t \in \mathcal{X}_t} exp(z_t^j)} ,$$
$$c_k^j = \lambda c_k^{j-1} + (1-\lambda) c_k^j , \tag{5}$$

where $c_k^j$ is the $k$-th centroid of the $j$-steps explorations, and $exp(\cdot)$ is the exponential function, $\lambda$ is the coefficient of movement. Then we estimate the label of each sample by its nearest initial centroid as $\hat{\mathcal{Y}}_t = \{\hat{y}_t^i | \hat{y}_t^i = \arg\min_k D(f_t^i, c_k^i)\}$, where the $D(\cdot, \cdot)$ is the cosine distance. While the $k$-th centroid is further modified by $c_k^j = \frac{\sum_{x_t \in \mathcal{X}_t} \mathbb{1}(\hat{y}_t^j = k) f_t^j}{\sum_{x_t \in \mathcal{X}_t} \mathbb{1}(\hat{y}_t^j = k)}$, where $\mathbb{1}$ is the indicator function, and the pseudo-labels are continuously updated in $j$-step iterations.

### 2.3.2 CONSISTENT LABELS EVALUATE EASY SAMPLES

With the pseudo-label evaluation strategy, we can obtain two individual pseudo-label sets as $\{\hat{\mathcal{Y}}_g, \hat{\mathcal{Y}}_c\}$, where the $\hat{\mathcal{Y}}_g$ and $\hat{\mathcal{Y}}_c$ are the label lists obtained from the ADM block $\mathcal{G}$ and source classifier $\mathcal{C}$, respectively. These two pseudo-label sets come from different dimensions of the feature representation. Then, we treat the features mapped by consistent labels as invariant features. Therefore, to define the categories of invariant features, we use the consistent pseudo-labels in the two spaces as the final pseudo-labels of the samples by $\hat{\mathcal{Y}}_e = \mathbb{1} \cdot \{\hat{\mathcal{Y}}_g, \hat{\mathcal{Y}}_c\}$, where $\mathbb{1} \in \{0, 1\}$ is an indicator function that returns 1 if $\hat{y}_g^i = \hat{y}_c^i$ evaluates as true. While the rest inconsistent set can be denote as $\hat{\mathcal{Y}}_h = \hat{\mathcal{Y}}_c \setminus \hat{\mathcal{Y}}_e$, where the $\hat{\mathcal{Y}}_h$ indicates the inconsistent hard labels. Then, we store the corresponding feature representations by the evaluated labels to easy sample bank $\mathcal{M}_e$ and hard sample bank $\mathcal{M}_h$. We argue that the sample discriminated by both domains is an easy sample that is more related to the source, i.e. source-like samples, while the rest samples are hard to discriminate or easily confused samples, which means they are closer to the target-specific situation. Therefore, we apply further treatment to them in the following.

### 2.3.3 CONSISTENT NEIGHBORS RE-EVALUATE HARD SAMPLES

As shown in Fig 1 (a), although there are significant differences between the hard samples and the source domain, a certain similarity is still between hard samples and easy samples. To further

discriminate the inconsistent samples, we follow such a principle that samples with higher similarity have closer clustering distributions in the common space. Therefore, we exploit the rating matrix to measure the similarity scores of the same space to vote the consistent labels. Concretely, we divide the feature representations $f_t$ into easy sample representations $f_e$ and hard feature representations $f_h$ by marked consistent labels. Then, we calculate the rating matrix of the divided samples in common space by $\mathcal{S}_{n \times m} = \{f_h \cdot f_e^\top\}$, where the $\mathcal{S}_{n \times m}$ indicates the similarity score of the features while $n$ and $m$ are count of corresponding feature representations s.t. $n + m = n_t$. For each inconsistent label $\hat{y}_h \in \hat{\mathcal{Y}}_h$, we exploit consistent label of its sampled $k$-nearest neighbors as $\tilde{y}_e^k = \hat{\mathcal{Y}}_e^{\arg \max_k \mathcal{S}}$, where $k$ is the index of the nearest neighbors and s.t. $k < m$. The re-evaluated hard labels are obtained by $\hat{y}_h = \hat{y}_e^k$. Then we combine the reconstructed pseudo-labels with the consistent labels to obtain the final pseudo-labels $\tilde{y}_t \in \tilde{\mathcal{Y}}_t$.

### 2.3.4 CONSISTENT SELF-SUPERVISED TRAINING

With the above strategies, we can obtain high-confidence pseudo-labels, which will benefit the training of self-supervised processes. The self-supervised loss is combined with two items as follows:

$$\mathcal{L}_{cst}^t = \frac{1}{n_b} \left( \sum_{i=1}^{n_b} \mathrm{CE}(\hat{z}_t^i, \tilde{y}_t^i) + \sum_{i=1}^{n_b} \mathrm{CE}(z_t^i, \tilde{y}_t^i) \right), \tag{6}$$

where $n_b$ is batch-size of each mini-batch. The first item of $CE(\cdot, \cdot)$ is used to optimize the auxiliary domain of $\mathcal{G}$, while the second item is used to optimize the target domain of $\mathcal{C}$. By self-supervised training, this loss can align the samples of the same label in different spaces, improving consistency discrimination from different dimensions.

### 2.4 ALIGN THE TARGET-SPECIFIC TO SOURCE-LIKE

Commonly in the SFDA setting, the source data cannot be accessed. Since the above consistent strategies are being exploited, we can discriminate the easy and hard samples from the target domain. From another perspective, easy samples are closer to the source domain, while hard samples are specific to the target domain. Therefore, we approximate easy samples as the source domain, i.e., $\mathcal{X}_e \sim \tilde{\mathcal{X}}_s$ and hard samples as the new target domain, i.e., $\mathcal{X}_h \sim \tilde{\mathcal{X}}_t$. All we need is to align the new target domain with the source domain. To solve the domain shift and encourage the feature representation to be invariant across different domains, traditional methods adopt different ways to align the differences between the two domains Long et al. (2015); Kang et al. (2019); Flamary et al. (2016). Among them, the kernel-based maximum mean discrepancy (MMD) Long et al. (2015) is widely used, which needs to meet the premise that the source domain and target domain can be accessed.

Assume that $\tilde{x}_s^i \in \tilde{\mathcal{X}}_s$ and $\tilde{x}_t^j \in \tilde{\mathcal{X}}_t$ are $i.i.d.$, the corresponding representations are sampled from the easy and hard samples $\mathcal{M}_e$ and $\mathcal{M}_h$ as $\tilde{f}_s^i$ and $\tilde{f}_t^j$, where $i \in n$, $j \in m$, and $n + m = n_t$. The motivation for MMD is that if two distributions are the same, all their statistics should be the same. In general, MMD defines the discrepancy between two distributions and their mean embedding in the reproducing kernel Hilbert space (RKHS) as $D_{\mathcal{H}}(\tilde{f}_s, \tilde{f}_t) := \sup_{\phi \sim \mathcal{H}} (\mathbb{E}_{\tilde{\mathcal{X}}_s}[\phi(\mathcal{F}_t(\tilde{\mathcal{X}}_s))] - \mathbb{E}_{\tilde{\mathcal{X}}_t}[\phi(\mathcal{F}_t(\tilde{\mathcal{X}}_t))])_{\mathcal{H}}$, where $\mathcal{H}$ is a function type.

To further obtain better domain adaptation, we adopt a new variant of MMD i.e., conditional guided multi-kernel MMD (CMK-MMD) to flexibly capture different features. For any two domains of feature distributions $\tilde{f}_s$ and $\tilde{f}_t$ with corresponding labels $\tilde{y}_s$ and $\tilde{y}_t$, we define the CMK-MMD between them as follows:

$$D_K^2(\zeta_s^i, \zeta_t^j) := \left\| \mathbb{E}_{\tilde{f}_s^i} \left[ \phi \left( \tilde{f}_s^i | \tilde{y}_s^i \right) \right] - \mathbb{E}_{\tilde{f}_t^j} \left[ \phi \left( \tilde{f}_t^j | \tilde{y}_t^j \right) \right] \right\|_{\mathcal{H}}^2, \tag{7}$$

where $D_k^2(\zeta_s^i, \zeta_t^j) = 0$ iff $\tilde{f}_s^i | \tilde{y}_s^j = \tilde{f}_t^j | \tilde{y}_t^j$. The kernel associated with the feature maps $\phi$, i.e., $K(\zeta_s, \zeta_t) = \langle \phi(\tilde{f}_s, \tilde{y}_s), \phi(\tilde{f}_t, \tilde{y}_t) \rangle$, Based on this we can explicit the multi-kernels defined as follows:

$$\mathcal{K} := \left\{ K = \sum_{u=1}^{v} \gamma_u K_u : \gamma_u \geq 0, \forall u \right\} \tag{8}$$

where $v$ is the number of kernels, $\gamma_u$ is the weight coefficients used to constrain the kernel function, and $\mathcal{K}$ is the multi-kernels, which leverage the different kernels to enhance CMK-MMD. Therefore, for any feature representation of two class $\tilde{f}_s^i$, $\tilde{f}_t^j$. CMK-MMD measures the discrepancy between two feature representations of source-like and target-specific, which is defined as follows:

$$\mathcal{L}_{\text{cmk}}^t = \frac{1}{n^2} \sum_{i=1}^n \sum_{j=1}^n \mathcal{K}\left(\zeta_s^i, \zeta_s^i\right) + \frac{1}{m^2} \sum_{i=1}^m \sum_{j=1}^m \mathcal{K}\left(\zeta_t^j, \zeta_t^j\right) - \frac{2}{nm} \sum_{i=1}^n \sum_{j=1}^m \mathcal{K}\left(\zeta_s^i, \zeta_t^j\right). \tag{9}$$

It is worth mentioning that although MMD has been explored in several papers Long et al. (2015); Kang et al. (2019); Zhang et al. (2022), to date there has been no attempt to enhance the hard feature representation via CMK-MMD in ViT-based architecture SFDA from the aspect of easy and hard samples. Besides, to make the target outputs individually certain and globally diverse, the common practice is to jointly perform feature learning, domain adaptation, and classifier learning by optimizing the information maximization Liang et al. (2020) loss function as follows:

$$\mathcal{L}_{im} = \mathcal{L}_{ent}(\mathcal{F} \circ \mathcal{C}; \mathcal{X}_t) + \mathcal{L}_{div}(\mathcal{F} \circ \mathcal{C}; \mathcal{X}_t) \tag{10}$$

where $\mathcal{L}_{ent}$ means entropy loss to explore the possible samples while $\mathcal{L}_{div}$ denotes the diverse loss to avoid the same one-hot encoding. With the above approaches, we can achieve the final total object loss:

$$\mathcal{L}_{total} = \mathcal{L}_{im} + \alpha \mathcal{L}_{cst} + \beta \mathcal{L}_{cmk}, \tag{11}$$

where $\alpha$ and $\beta$ are the balance factor to control each loss item.

## 3 EXPERIMENTS

### 3.1 DATASETS AND IMPLEMENTATION DETAILS

**Office-31** Saenko et al. (2010) is a standard small-sized DA benchmark, which contains 4,652 images with 31 classes from three domains (Amazon (A), DSLR (D), and Webcam (W)). **Office-Home** Venkateswara et al. (2017) is a challenging medium-sized benchmark, which consists of 15,500 images with 65 classes from four domains (Artistic images (Ar), Clip Art (Cl), Product images (Pr), and Real-World images (Rw)). **VisDA-C** Peng et al. (2017) is a challenging large-scale Synthetic-to-Real dataset that focuses on the 12-class object recognition task. The source domain contains 152 thousand synthetic images generated by rendering 3D models while the target domain has 55 thousand real object images sampled from Microsoft COCO.

Our method is based on the ViT-base/16 and Deit-base/16 [1], pre-trained on ImageNet-1k. The input size of the image in our experiments is $224 \times 224$, and each image is split into 16 patches. Besides, the head account in each layer of our backbone is 12. The learning rate for Office-31 and Office-Home is set to 1e-2, while for VisDA-C is 1e-3. The training epoch is set to 100 and the whole process is optimized by stochastic gradient descent (SGD) with a momentum of 0.9 and weight decay 1e-3. The batch size is 64 by default. The trade-off parameters $\alpha$, $\beta$, and $\tau$ are set as 0.3, 0.1, and 0.07, respectively. All experiments are conducted with PyTorch on NVIDIA 3090 GPUs.

### 3.2 BASELINES AND COMPARISON METHODS

We compare the proposed approach with the state-of-the-art methods mainly under the closed-set unsupervised SFDA. For the closed-set DA, the compared methods include source-free methods of ResNet-50 and ResNet-101 backbones: SFDA Kim et al. (2021), SHOT Liang et al. (2020), 3C-GAN Li et al. (2020), A$^2$Net Xia et al. (2021), G-SFDA Yang et al. (2021b), SFDA-DE Ding et al. (2022), TransDA Yang et al. (2021a), and our source-free methods of ViT backbone: Source-only, ViT-SHOT, ViT-base, Deit-SHOT, and Deit-base.

Our ViT-Base and Deit-base backbones are compared with the ResNet-50 He et al. (2016) backbone on small-sized and medium-sized datasets, i.e. Office-31 and Office-Home. For the sake of fairness, we compare with ResNet-101 on the large-scale dataset, i.e. VisDA-C. For each benchmark, we exploit the self-supervised training loss $\mathcal{L}_{self}$ as our baseline. 'Source-only' represents exploiting the whole source model for target label prediction without domain adaptation.

---

[1] https://github.com/huggingface/pytorch-image-models/tree/main/timm

Table 1: Accuracy (%) of different methods of unsupervised SFDA in Office-31 dataset.

| Method | S.F. | A→D | A→W | D→A | D→W | W→A | W→D | Avg. |
|---|---|---|---|---|---|---|---|---|
| ResNet-50 backbone | | | | | | | | |
| Source-only He et al. (2016) | − | 68.4 | 96.7 | 99.3 | 68.9 | 62.5 | 60.7 | 76.1 |
| SHOT Liang et al. (2020) | ✓ | 94.0 | 90.1 | 74.7 | 98.4 | 74.3 | 99.9 | 88.6 |
| SHOT++ Liang et al. (2021) | ✓ | 94.3 | 90.4 | 76.2 | 98.7 | 75.8 | 99.9 | 89.2 |
| 3C-GAN Li et al. (2020) | ✓ | 92.7 | 93.7 | 75.3 | 98.5 | 77.8 | 99.8 | 89.6 |
| A²Net Xia et al. (2021) | ✓ | 94.5 | 94.0 | 76.7 | 99.2 | 76.1 | 100 | 90.1 |
| SFDA-DE Ding et al. (2022) | ✓ | 96.0 | 94.2 | 76.6 | 98.5 | 75.5 | 99.8 | 90.1 |
| TransDA Yang et al. (2021a) | ✓ | 97.2 | 95.0 | 73.7 | 99.3 | 79.3 | 99.6 | 90.7 |
| ViT backbone | | | | | | | | |
| Source-only | − | 88.0 | 90.1 | 71.5 | 95.2 | 73.4 | 99.2 | 86.2 |
| G-SFDA (ViT-base)* | ✓ | 95.6 | 96.4 | 79.6 | 98.1 | 78.3 | 99.6 | 91.3 |
| SHOT++ (ViT-base)* | ✓ | 96.2 | 95.7 | 79.6 | 99.0 | 80.7 | 99.8 | 91.8 |
| **AudoFormer (ViT-base)** | ✓ | 97.8 | 97.6 | 81.7 | 99.3 | 81.7 | 99.8 | **93.0** |
| DeiT backbone | | | | | | | | |
| Source-only | − | 88.8 | 88.9 | 70.4 | 96.6 | 74.0 | 99.0 | 86.3 |
| G-SFDA (DeiT-base)* | ✓ | 95.6 | 96.4 | 79.6 | 98.1 | 78.3 | 99.6 | 91.3 |
| SHOT++ ( DeiT-base)* | ✓ | 96.4 | 97.2 | 80.8 | 98.7 | 79.8 | 99.6 | 92.1 |
| **AudoFormer (DeiT-base)** | ✓ | 97.6 | 96.9 | 83.0 | 99.1 | 82.8 | 99.8 | **93.2** |

The results marked by '∗' come from our re-implementations. The rows of gray background are the best performance.

### 3.2.1 RESULTS OF OFFICE-31

As can be seen from Table 1, For the ResNet-50 approaches in source-free settings, such as SHOT, SHOT++, 3C-GAN, A²Net, SFDA-DE, and TransDA, the best performance of these approaches can achieve around 90%, which demonstrates SFDA can achieve good performance even in the scenario without source data. At the bottom of Table 1 are our AudoFormer, a variant of ViT-base and DeiT-base models, which significantly outperform previously published state-of-the-art approaches, advancing the average accuracy from 93.0% and 93.2%. Obviously, in the source-free setting, AudoFormer can outperform those methods based on the ResNet backbones. For a fair comparison, we also implemented the state-of-the-art approaches of pseudo-label evaluation (i.e., SHOT++) and consistent neighbor (i.e., G-SFDA) on both ViT and DeiT backbones. From the table, our method achieved improvements of 1.2 and 1.1 percentage points, respectively.

Table 2: Accuracy (%) of different methods of unsupervised SFDA on the Office-Home dataset.

| Method | S.F. | Ar→Cl | Ar→Pr | Ar→Rw | Cl→Ar | Cl→Pr | Cl→Rw | Pr→Ar | Pr→Cl | Pr→Rw | Rw→Ar | Rw→Cl | Rw→Pr | Avg. |
|---|---|---|---|---|---|---|---|---|---|---|---|---|---|---|
| ResNet-50 backbone | | | | | | | | | | | | | | |
| Source-only He et al. (2016) | − | 34.9 | 50.0 | 58.0 | 37.4 | 41.9 | 46.2 | 38.5 | 31.2 | 60.4 | 53.9 | 41.2 | 59.9 | 46.1 |
| SHOT Liang et al. (2020) | ✓ | 57.1 | 78.1 | 81.5 | 68.0 | 78.2 | 78.1 | 67.4 | 54.9 | 82.2 | 73.3 | 58.8 | 84.3 | 71.8 |
| G-SFDA Yang et al. (2021b) | ✓ | 57.9 | 78.6 | 81.0 | 66.7 | 77.2 | 77.2 | 65.6 | 56.0 | 82.2 | 72.0 | 57.8 | 83.4 | 71.3 |
| A²Net Xia et al. (2021) | ✓ | 58.4 | 79.0 | 82.4 | 67.5 | 79.3 | 78.9 | 68.0 | 56.2 | 82.9 | 74.1 | 60.5 | 85.0 | 72.8 |
| SFDA-DE Ding et al. (2022) | ✓ | 59.7 | 79.5 | 82.4 | 69.7 | 78.6 | 79.2 | 66.1 | 57.2 | 82.6 | 73.9 | 60.8 | 85.5 | 72.9 |
| SHOT++ Liang et al. (2021) | ✓ | 57.9 | 79.7 | 82.5 | 68.5 | 79.6 | 79.3 | 68.5 | 57.0 | 83.0 | 73.7 | 60.7 | 84.9 | 73.0 |
| TransDA Yang et al. (2021a) | ✓ | 67.5 | 83.3 | 85.9 | 74.0 | 83.8 | 84.4 | 77.0 | 68.0 | 87.0 | 80.5 | 69.9 | 90.0 | 79.3 |
| ViT backbone | | | | | | | | | | | | | | |
| Source-only | − | 51.4 | 81.1 | 85.4 | 73.6 | 82.2 | 83.0 | 73.6 | 50.8 | 87.2 | 78.2 | 50.1 | 86.4 | 73.6 |
| G-SFDA (ViT-base)* | ✓ | 63.9 | 81.0 | 84.5 | 73.3 | 82.5 | 82.5 | 73.6 | 62.6 | 85.1 | 78.1 | 64.7 | 87.2 | 76.6 |
| SHOT++ (ViT-base)* | ✓ | 68.7 | 87.2 | 88.4 | 79.0 | 87.7 | 93.1 | 78.8 | 59.0 | 89.7 | 81.3 | 59.2 | 90.1 | 80.2 |
| **AudoFormer (ViT-base)** | ✓ | 64.3 | 87.8 | 86.3 | 82.5 | 91.8 | 88.4 | 80.3 | 60.8 | 88.9 | 82.5 | 76.8 | 90.5 | **81.7** |
| DeiT backbone | | | | | | | | | | | | | | |
| Source-only | − | 56.7 | 77.3 | 83.0 | 68.3 | 74.6 | 78.0 | 67.2 | 53.5 | 82.8 | 74.0 | 55.4 | 84.0 | 71.2 |
| G-SFDA (Deit-base)* | ✓ | 65.1 | 82.2 | 85.6 | 73.6 | 80.4 | 82.0 | 75.4 | 62.7 | 85.3 | 79.0 | 65.4 | 87.5 | 77.0 |
| SHOT++ (Deit-base)* | ✓ | 60.3 | 86.6 | 87.5 | 82.0 | 89.8 | 89.0 | 80.7 | 62.1 | 89.6 | 82.6 | 77.1 | 90.5 | 81.5 |
| **AudoFormer (DeiT-base)** | ✓ | 70.3 | 88.7 | 90.0 | 83.1 | 90.2 | 89.9 | 80.1 | 62.9 | 90.5 | 81.8 | 74.3 | 92.5 | **82.9** |

### 3.2.2 RESULTS OF OFFICE-HOME

According to the results on the Office-Home dataset shown in Table 2, exploiting the ViT-backbone and DeiT-backbone directly to evaluate the target domain benchmark before adaptation (i.e., Source-only) can obtain a score of 73.6% and 71.2% respectively, while that of the ResNet-50 backbone is only 46.1%. The best source-free result is 79.3% gotten by TransDA. In comparison, the performance of our AudoFermor is 81.7% in the source-free setting, surpassing the TransDA by 2.4%, which proves the necessity of exploiting the consistent strategy of global attention in domain adaptation. For the TransDA approach, although it combines the advantages of ResNet and Transformer, our performance is still better than its performance in terms of average. Besides, in the same setting, the SHOT++ and G-SFDA exploiting the ViT-base backbone can achieve an average score of 76.6% and 80.2%. While the best performance of our AudoFormer can achieve a score of 81.7. Likewise,

when exploiting the DeiT-backbone, the variant AudoFormer can also outperform the state-of-the-art by 1.4%, demonstrating its effectiveness.

### 3.2.3 RESULTS OF VISDA-C

As shown in Table 3, we mainly compare the results of SFDA on the ResNet-101 backbones due to its super performance. First, we notice that the source model with the ViT backbone surpasses that with the ResNet-101 backbone by 27.5%. This shows the superiority of vision transformers on DA. Therefore, for CNN-based methods, a more rational design is needed. The best performance of source-free ResNet-101 approach SHOT++ on the VisDA-C dataset achieves an average accuracy of 87.3%, which outperforms the other result of SFDA-DE and G-SFDA. However, the performance of our AudoFormer both exploiting Deit-base and ViT-base can easily outperform those methods of ResNet-101 backbone. For the SHOT++, if directly exploiting the settings of ViT-base and DeiT-base, they can achieve a limited improvement, e.g., for both settings, their combination only can improve the baseline by 0.3 and 0.6, respectively. When exploiting the AudoFormer, our approach can improve the SHOT++ by 0.5 and 0.9 respectively, showing its performance. Therefore, we argue that our approach can extract additional information from the auxiliary domain to compensate for the invariable feature representation while our consistency strategy based on the auxiliary domain can provide some improvement for the progress of domain adaptation.

Table 3: Accuracy (%) of different methods of unsupervised SFDA in VisDA-C dataset.

| Method | S.F. | plane | bicycle | bus | car | horse | knife | motor | person | plant | skate | train | truck | Avg. |
|---|---|---|---|---|---|---|---|---|---|---|---|---|---|---|
| ResNet-101 backbone | | | | | | | | | | | | | | |
| Source-only He et al. (2016) | − | 55.1 | 53.3 | 61.9 | 59.1 | 80.6 | 17.9 | 79.7 | 31.2 | 81.0 | 26.5 | 73.5 | 8.5 | 52.4 |
| SFDA Kim et al. (2021) | ✓ | 86.9 | 81.7 | 84.6 | 63.9 | 93.1 | 91.4 | 86.6 | 71.9 | 84.5 | 58.2 | 74.5 | 42.7 | 76.7 |
| 3C-GAN Li et al. (2020) | ✓ | 94.8 | 73.4 | 68.8 | 74.8 | 93.1 | 95.4 | 88.6 | 84.7 | 89.1 | 84.7 | 83.5 | 48.1 | 81.6 |
| SHOT Liang et al. (2020) | ✓ | 94.3 | 88.5 | 80.1 | 57.3 | 93.1 | 94.9 | 80.7 | 80.3 | 91.5 | 89.1 | 86.3 | 58.2 | 82.9 |
| TransDA Yang et al. (2021a) | ✓ | 97.2 | 91.1 | 81.0 | 57.5 | 95.3 | 93.3 | 82.7 | 67.2 | 92.0 | 91.8 | 92.5 | 54.7 | 83.0 |
| A$^2$Net Xia et al. (2021) | ✓ | 94.0 | 87.8 | 85.6 | 66.8 | 93.7 | 95.1 | 85.8 | 81.2 | 91.6 | 88.2 | 86.5 | 56.0 | 84.3 |
| G-SFDA Yang et al. (2021b) | ✓ | 96.1 | 88.3 | 85.5 | 74.1 | 97.1 | 95.4 | 89.5 | 79.4 | 95.4 | 92.9 | 89.1 | 42.6 | 85.4 |
| SFDA-DE Ding et al. (2022) | ✓ | 95.3 | 91.2 | 77.5 | 72.1 | 95.7 | 97.8 | 85.5 | 86.1 | 95.5 | 93.0 | 86.3 | 61.6 | 86.5 |
| SHOT++ Liang et al. (2021) | ✓ | 97.7 | 88.4 | 90.2 | 86.3 | 97.9 | 98.6 | 92.9 | 84.1 | 97.1 | 92.2 | 93.6 | 28.8 | 87.3 |
| ViT backbone | | | | | | | | | | | | | | |
| Source-only | − | 96.7 | 69.8 | 74.7 | 76.0 | 86.0 | 28.2 | 82.7 | 16.5 | 63.9 | 90.0 | 93.4 | 14.7 | 66.1 |
| G-SFDA (ViT-base)* | ✓ | 96.5 | 93.7 | 85.9 | 74.9 | 96.4 | 97.0 | 86.0 | 61.1 | 89.6 | 96.3 | 94.3 | 58.9 | 85.8 |
| SHOT++ (ViT-base)* | ✓ | 97.3 | 93.4 | 86.4 | 76.6 | 96.2 | 96.9 | 85.2 | 88.0 | 93.1 | 89.1 | 92.1 | 57.7 | 87.6 |
| **AudoFormer (ViT-base)** | ✓ | 97.5 | 93.4 | 87.4 | 76.8 | 95.8 | 97.1 | 85.7 | 88.8 | 94.6 | 88.8 | 88.1 | 60.5 | **87.8** |
| DeiT backbone | | | | | | | | | | | | | | |
| Source-only | − | 97.7 | 36.2 | 84.3 | 61.7 | 69.2 | 45.3 | 95.3 | 10.4 | 76.7 | 41.1 | 93.7 | 29.1 | 61.7 |
| G-SFDA (DeiT-base)* | ✓ | 98.1 | 87.1 | 88.0 | 70.3 | 95.7 | 98.9 | 93.9 | 67.8 | 94.9 | 83.5 | 94.5 | 65.9 | 86.6 |
| SHOT++ (DeiT-base)* | ✓ | 98.3 | 88.7 | 88.2 | 71.3 | 96.1 | 98.1 | 94.9 | 83.0 | 94.9 | 88.1 | 94.6 | 58.4 | 87.9 |
| **AudoFormer (DeiT-base)** | ✓ | 98.9 | 86.6 | 88.6 | 72.3 | 98.1 | 98.1 | 95.9 | 83.0 | 94.1 | 98.1 | 94.6 | 50.4 | **88.2** |

### 3.3 CONCLUSION

In this paper, we propose a novel approach AudoFormer with a consistent auxiliary domain. The main innovations of AudoFormer are attributed to three technical components: auxiliary domain, consistency strategies, and CMK-MMD. AudoFormer with an auxiliary domain can achieve auxiliary discriminative representations, while multiple consistent strategies can dynamically evaluate the pseudo labels for self-supervised learning and distinguish source-like easy samples and target-specific hard samples for SFDA. Besides, the CMK-MMD is exploited to align the source-like domain and target-specific domain. The experimental results demonstrate the effectiveness of our proposed approaches.

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
