# OpenReview forum: "AudoFormer: An Efficient Transformer with Consistent Auxiliary Domain for Source-free Domain Adaptation"
_ICLR.cc/2024/Conference — ICLR 2024 Conference Withdrawn Submission_

### Official Review · Reviewer_n97E · 2023-10-22

**Soundness:** 3 good
**Presentation:** 3 good
**Contribution:** 2 fair
**Rating:** 5
**Confidence:** 2

**Summary:**

Aiming to solve the domain generalization problem in the case of inaccessible source domains, the authors propose AudoFormer which dynamically evaluates consistent labels and consistent neighbors through ADM blocks. and realizes sample alignment using CMK- MMD

**Strengths:**

Uses VIT instead of CNN as backbone, fewer previous studies have used VIT for this purpose.
Score the samples by multiple consistency strategies to further categorize them into simple and difficult samples.
Use CM K- MMD to align difficult samples with simple samples.

**Weaknesses:**

Not innovative enough, it's all stuff that's already been proposed before, and only one alignment module is self-proposed.
Lack of ablation experiments to analyze the effect of the three modules proposed by ourselves

**Questions:**

Is it possible to add a proof-of-validity analysis of the three modules in the ablation experiment

---

> ### Author Response · Authors · 2023-11-16
> **About novelty and ablation experiment**
>
> We apologize for any confusion caused.
> 1). About novelty. First, most current frameworks of SFDA are mainly based on the traditional architecture, i.e., Resnet series, and very few studies use the Transformer as the backbone for the research. We propose a variant of Transformer with an auxiliary domain that alleviates the dilemma of SFDA without a source domain to some extent. We solve this task from a different aspect by domain consistency which is first proposed in our work and is different from other approaches. Besides, we exploit the MMD with conditional guided information (i.e., CMK-MMD) to improve the effect of the alignment. By Tab 5 in Appendix B, CMK-MMD can further improve the effect of the experiment.
> 2). About the ablation experiment. The ablation studies about the effect of the three modules are offered in Appendix B. Reviewers can check them in our attachments.

---

> > ### Comment · Reviewer_n97E · 2023-11-17
> > **Response to comments**
> >
> > First, regarding the ablation experiments, I'm very sorry I overlooked the experiments in the appendix, thanks for the reminder. Second, while I still don't see anything innovative in just switching the backbone from CNN to Transformer, perhaps you should focus your description more on the domain consistency perspective.

---

> > > ### Author Response · Authors · 2023-11-17
> > > **About innovation**
> > >
> > > Thanks for your further feedback. We agree with the reviewer about the more description on the domain consistency perspective and we will continue to improve our paper. But about innovation, we want to make it more clear. Firstly, we're not just using transformer backbone. To address SFDA such a specific task, we are not only designing auxiliary domain module but also using some techniques such as EMA in the process of handling features and knowledge distillation in the first stage. In the process of domain adaptation, we adopt a dynamic way to determine the centroid which is different from the traditional centroid processing method in SHOT, SHOT++, G-SFDA, SFDA-DE, etc. Our approach of easy samples and hard samples by assessing consistency in both domains is also proposed for the first time. In addition, MMD is often used in scenarios where the source domain is accessible, which is difficult to apply directly on our SFDA. Because our method identifies the source-like samples through the consistency of the auxiliary domain, it can be aligned through the MMD. To further improve performance, we also introduced conditional information for MMD (i.e., CMK-MMD).

---

> ### Author Response · Authors · 2023-11-22
>
> Dear reviewer. I kindly inquire whether our response has successfully addressed your uncertainties. We remain dedicated to addressing concerns you may possess with utmost eagerness. Additionally, we would be encouraged if the reviewer considers raising the score.

---

### Official Review · Reviewer_2ckR · 2023-10-29

**Soundness:** 3 good
**Presentation:** 3 good
**Contribution:** 3 good
**Rating:** 5
**Confidence:** 5

**Summary:**

This paper proposes a method for source-free domain adaptation. They utilize the Transformer model as the backbone for the training. In addition, the intermediate layer features are aggregated and considered as auxiliary domain representations. They then align the source-like with the target-specific samples by conditional guided multi-kernel max mean discrepancy (CMK-MMD), which guides the hard samples to align the corresponding easy samples.  Some experimental evaluation validates the good results of the method in source-free domain adaptation.

**Strengths:**

1. The paper's writing is generally sound, with clear expressions.
2. The proposed methods are interesting, which consider the intermediate layers as a kind of representation of the auxiliary domain, and try to align them to the target domain.
3. The pseudo-label processing technique is novel, which plays a critical role in alignment of different domains.

**Weaknesses:**

1. The paper's organization confuses me sometimes, for example, 2.3.2 and 2.2.3 should be in the same section, perhaps.
2.  There is a critical weakness of this paper, I have not found the ablation studies for each component/strategy of the proposed method, which makes it difficult to evaluate how strong these methods are.
3. I guess, the alignment with pseudo-labels is strong enough, is it possible to say that MMD alignment is not necessary?

**Questions:**

1. I suggest the paper add a thorough analysis of each component, and ablation studies should be included.
2. Qualitative evaluation is needed.
3. Which one is stronger? MMD alignment or pseudo-label-based alignment?

---

> ### Author Response · Authors · 2023-11-16
> **About paper's organization**
>
> Thank you for your comments, if I understand you correctly, 2.3.2 and 2.3.3 should be in the same section, cause no section 2.2.3 in our article.
> In 2.3, we exploit multiple consistency strategies, i.e., first dynamically evaluating the centroids of categories, second evaluating the easy samples by consistent labels, then evaluating the hard samples by consistent neighbors, and at last, training the model by consistent self-supervision. Thus, these strategies are juxtaposed in importance and progressive in process.

---

> ### Author Response · Authors · 2023-11-16
> **About ablation studies**
>
> In Appendix B of supplementary material, we provide ablation studies about the performance of ADM component on source domains. With Tab 4, we can find that AudoFormer can effectively enhance supervised learning and domain adaptation on source domains. For main strategies proposed in our paper, we show its effect in Tab 5.

---

> ### Author Response · Authors · 2023-11-16
> **About pseudo-labels and MMD**
>
> MMD can align the feature embedding of hard samples to easy samples in the reproducing kernel Hilbert space (RKHS), which is a different way from supervised learning of pseudo-labels in common space. According to the experimental results in Table 5 of Appendix B, CMK-MMD can further improve the effect of the experiment. Therefore, a combination of these two approaches is beneficial in enhancing the effectiveness of the model.

---

> ### Author Response · Authors · 2023-11-22
>
> Dear reviewer. I kindly inquire whether our response has successfully addressed your uncertainties. We remain dedicated to addressing concerns you may possess with utmost eagerness. Additionally, we would be encouraged if the reviewer considers raising the score.

---

### Official Review · Reviewer_QH22 · 2023-10-29

**Soundness:** 3 good
**Presentation:** 2 fair
**Contribution:** 2 fair
**Rating:** 5
**Confidence:** 4

**Summary:**

This paper introduces a model called AudoFormer to address the issue of obtaining invariant feature representations by domain consistency in SFDA. In the pre-training phase, the model employs a Visual Transformer (ViT) as the backbone and trains an Auxiliary Domain Module (ADM) based on the global attention features from the intermediate layers of the feature extractor to generate diverse representations. During the domain adaptation phase, this paper utilizes a consistency strategy to categorize target samples into "source-like" easy samples and "target-specific" hard samples, based on both the auxiliary domain and the target domain. It then optimizes their pseudo-labels to reduce the impact of noise. Finally, it aligns the hard samples with their corresponding easy samples using CMK-MMD. Experiments are conducted on three datasets, i.e., Office-31, Office-Home, and VISDA-C, to show the effectiveness of the proposed method.

**Strengths:**

#Originality
This paper introduces an Auxiliary Domain Module (ADM) block for the ViT backbone, addressing the inherent limitations of inductive bias and enabling the generation of diverse representations from global attention. These diverse representations are used to construct an auxiliary domain. Subsequently, the approach treats features mapped by consistent labels as invariant features, effectively tackling one of the most challenging issues in SFDA. Additionally, the paper leverages a dynamic strategy to calculate the initial centroid of each category, thereby mitigating the interference caused by noise. Furthermore, the self-supervised loss is applied to align samples with the same label in different spaces, enhancing the consistency discrimination from multiple dimensions. To achieve even better domain adaptation, this paper introduces CMK-MMD. This variant enhances the hard feature representation.

#Quality
This paper conducts experiments on benchmarks of varying sizes. In all three sets of experiments, the proposed method in this paper outperforms other experiments listed. Furthermore, the supplementary documentation includes an ablation study on the ADM module and the consistency strategy, demonstrating their effectiveness in improving performance. The paper also provides visual insights by employing attention maps and t-SNE to visualize various methods, substantiating the efficacy of the proposed approach.

#Clarity
The clarity of this paper is relatively high.

#Significance
This paper introduces a novel approach that applies Transformer-based methods to the challenging problem of Source-Free Domain Adaptation (SFDA). By leveraging an Auxiliary Domain Module, it effectively mitigates the impact of inductive bias, overcomes limitations imposed by the convolutional neural network's receptive field, and preserves both global and local features. This, in turn, enhances the model's ability to extract semantic information. Furthermore, the paper presents a methodology rooted in the consistency principle between the auxiliary domain and the target domain, enabling the extraction of invariant features.

**Weaknesses:**

Experiment: It would be beneficial to include comparisons with more recent models. As of 2023, recent papers have demonstrated an average accuracy of around 90.0 on the VisDA-C dataset. However, most of the methods compared in this paper are from 2021 or earlier, with only a few from 2022. This leaves the paper lagging behind in terms of benchmarking against the most up-to-date approaches. Additionally, it would be worthwhile to provide a more comprehensive exploration of the effectiveness of the improvements made to MMD and the centroid evaluation calculation methods. Demonstrating the impact of these enhancements on the experimental results would further strengthen the paper.
An experiment on Domain-Net, one of the largest DA datasets, is required but missing.

Innovation: The optimization methods for pseudo-labels and the techniques for reducing distribution loss are fairly standard and well-established. While there are some modifications introduced in the paper, their effectiveness hasn't been convincingly demonstrated.

Details: There are several issues with the visual representation in the paper. On the third page, the color for "category centroid" and "category dynamic centroid" in the overall workflow diagram of AudoFormer is not very intuitive. Additionally, the arrows between the "consistency strategies" module and the "align the target-specific to source-like" module seem to depict data flow direction, but the legend indicates that the left arrows represent the "back loss." On the second page, there is a typographical error in the third-to-last line, where "source-like" is misspelled. In section 3.2.2 on the eighth page, there is a spelling error in the term "AudoFormer." Furthermore, on the thirteenth page, the title of Table 4 contains a spelling error for "ADM."

Content: One notable aspect that could improve the paper is the inclusion of a more comprehensive introduction to the related work. By providing a thorough survey of existing research and methodologies in the same domain, the paper could offer readers a clearer understanding of where the proposed approach fits within the broader context of the field. This would not only enhance the paper's background but also help readers appreciate the novelty and significance of the presented work.

**Questions:**

Besides the weaknesses above, what are the advantages of using an auxiliary domain and consistency-based methods to distinguish between "source-like" and "target-specific" samples, compared to the traditional approach that relies solely on entropy levels for differentiation?



----------after rebuttal-----------

Thank the authors for providing the rebuttal. This paper received four "marginally below the acceptance threshold". The reviews have various concerns, such as motivation, novelty, and ablation study. I think the paper might not be suitable to be accepted in its current format.

---

> ### Author Response · Authors · 2023-11-16
> **About Experiment**
>
> At the time we conducted such a study, the most recent work on 2023 was not available yet. We thank the reviewers for the reminder and we will add relevant comparisons in revised versions.
> In our Appendix, we offered the analysis of the ADM and other three proposed components, we will further add some evaluation.
> We are in the process of adding related experiments about Domain-Net. Since it takes some time, once the results are available we will provide them to the reviewers.

---

> ### Author Response · Authors · 2023-11-16
> **About proof of validity**
>
> Regarding the proof of the effectiveness of the individual strategies presented in the paper, we have given in Appendix B. Reviewer can check our supplementary material.  In addition, to further validate the effect of our proposed dynamic centroids (Dyn. centroids) and the conventional static centroids (Sta. centroids). We also added the following experiments:
>
> |Datasets | Sta. centroids |Dyn. centroids |
> | :---: | :---: | :---: |
> | Office31 | 92.3 | 93.0  |
> | Office-Home | 80.9  | 81.7  |
> | VISDA-C | 87.2 | 87.8|
> | Ave. | 86.8 | 87.5|
>
> As can be seen from the Tab., we consider that Dyn. centroid can evaluate more reasonable center of the samples, and thus achieve better results on three datasets.

---

> ### Author Response · Authors · 2023-11-16
> **About Details**
>
> Thanks to the author's detailed comments, for "category centroid" and "category dynamic centroid", we will reset the colors in our revised version. Besides, the colors of data flow and loss are actually different, but for clarity, we will also make further changes. For the other typos in our paper, we will carefully review and revise in the revision. Thanks again to the reviewer.

---

> ### Author Response · Authors · 2023-11-16
> **About Content**
>
> About related work, we thank the reviewer for the valuable feedback and thoughtful comments. In the original version, our paper provided an introduction to related work. But we have made some selections in main paper due to space constraints. In the revised version, we will attach a comprehensive introduction of the relevant work.

---

> ### Author Response · Authors · 2023-11-16
> **About the advantages of using an auxiliary domain**
>
> Domain consistency is essentially the use of multi-domain consistency to judge invariance features, and our initial inspiration comes from the domain adaptation (DA) accessible to the source domain. Therefore, we construct the source domain in the way of an auxiliary domain and hope that it can play the role of source domain. Compared to the traditional approaches, our method has more feature information to use, and the category centroid with consistency discrimination has higher confidence, while those methods that rely solely on entropy levels are more susceptible to noise.

---

> ### Author Response · Authors · 2023-11-22
>
> Dear reviewer. I kindly inquire whether our response has successfully addressed your uncertainties. We remain dedicated to addressing concerns you may possess with utmost eagerness. Additionally, we would be encouraged if the reviewer considers raising the score.

---

### Official Review · Reviewer_JA6w · 2023-11-05

**Soundness:** 3 good
**Presentation:** 3 good
**Contribution:** 2 fair
**Rating:** 5
**Confidence:** 4

**Summary:**

This paper introduces AudoFormer, an efficient transformer-based model for SFDA, which leverages an auxiliary domain module to obtain diverse representations and employs consistency strategies to distinguish invariable features, ultimately achieving superior performance on benchmark datasets compared to existing methods.

**Strengths:**

- This paper first solves SFDA problem from a new perspective by domain consistency.
- This paper aligns the source-like with target-specific samples by CMK-MMD to improve the alignment effect of the domain adaptation.
- Extensive experiments are conducted on three benchmark datasets to show its SOTA performance.

**Weaknesses:**

- Motivation is not clear. The last two sentences of the first paragraph on page 2 ("Intuitively, if ...... layer features.") do not have a direct cause-and-effect relationship. Why should we turn to intermediate layer features for emulating the invariant features? And the "Intuitively" also lacks clear explanations.
- Lack of novelty. For instance, dividing target samples into easy and hard parts is proposed by [1], and consistency between neighborhoods is proposed by [2].
- The use of CMK-MMD is not clarified. There are lots of methods for constructing invariant feature representations across different domains, but no one is compared to used CMK-MMD.
- More experiments and results are needed. While ViT is a stronger backbone to ResNet, and SFDA-DE and TransDA can be equipped with ViT, why don't conduct experiments on it? Besides, recent commonly used DomainNet should be included, and more ablation studies are needed.

[1] Divide and Contrast: Source-free Domain Adaptation via Adaptive Contrastive Learning
[2] Exploring Domain-Invariant Parameters for Source-Free Domain Adaptation

**Questions:**

See the weakness above.

---

> ### Author Response · Authors · 2023-11-16
> **About Motivation**
>
> We apologize for any confusion caused. About the sentence of ("Intuitively, if ...... layer features.") ,  what we're trying to convey is this current mainstream approach is based on invariant features, however, these invariant features are relatively difficult to build in the SFDA.  The "Intuitively" is way of our hypotheses and judgment, which prompts the presentation of the methodology and subsequent experimental verification. We will carefully scrutinize our sentences in the revised version.

---

> ### Author Response · Authors · 2023-11-16
> **About novelty**
>
> First, we solve the SFDA problem by exploiting a variant of Transformer with an auxiliary domain that alleviates the dilemma of SFDA without a source domain to some extent.
> Second, we solve the invariable feature representation from a new perspective by domain consistency which hasn't been studied by anyone else.
> Besides, we exploit the MMD with conditional guided information (i.e., CMK-MMD) to further improve the effect of the alignment.
>
> In [1], DaC mainly exploits contrastive learning to generate robust class centroids, which is different from the way of our domain consistency. Consistent neighbors in our approach are just a strategy that is used to identify difficult samples, while in [2], it is the key approach to evaluate the centroids.

---

> ### Author Response · Authors · 2023-11-17
> **About CMK-MMD**
>
> First, MMD is widely used in DA tasks and has been proven to work well. Second, most approaches ignore conditional information which can also be used to guide learning. Thus, we proposed to exploit CMK-MMD such a variant. Besides, MMD is commonly used to align source and target domains in scenarios where the source domain is accessible. In the SFDA scenario, it is difficult to use MMD for alignment when the source domain is inaccessible. We can exploit this approach due to our auxiliary domain module can evaluate source-like samples. We are adding some comparative experiments that will be given soon.  Thanks for your reminder, we will further add descriptions of CMK-MMD in the revised version.

---

> ### Author Response · Authors · 2023-11-17
> **About experiments and comparison**
>
> First, SFDA-DE is not open source, we have tried to reproduce it, but the performance is not as good as announced, probably due to some unknown settings. Therefore, for fairness, our AudoFormer does not reimplement and compare with it.
> Second, TransDA itself is already based on the Transformer architecture, so there is no need to reuse our method to implement it.
> Besides, about DomainNet, similar to the setting of [1, 3, 4]，we also select four domains (i.e., Real, Clipart, Painting and Sketch) with 126 classes as the single-source unsupervised domain adaptation benchmark and construct seven single-source adaptation scenarios from the selected four domains. As can been from the following table, our approach is still able to achieve advantages on the large dataset DomainNet.
> | Method | Rw $\rightarrow$ Cl| Rw$\rightarrow$Pt| Pt$\rightarrow$Cl| Cl$\rightarrow$Sk|Sk$\rightarrow$Pt| Rw$\rightarrow$Sk| Pt$\rightarrow$Rw | Avg.|
> | :--- | :---: | :---: | :---: | :---: | :---: | :---: | :---: | :---: |
> | Source-only| 67.1 | 73.1 | 69.4 | 64.0 | 70.9 | 57.4 | 86.1 | 69.7 |
> | G-SFDA| 71.3 | 76.1 | 73.5 | 69.5 | 76.2 | 66.2 | 87.4 | 74.3 |
> | SHOT++| 73.4 | 76.0 | 74.2 | 70.5 | 77.4 | 66.0 | 88.8 | 75.2 |
> | AudoFormer (ViT-base)| 74.3 | 76.3 |75.6 | 73.5| 77.5 | 65.9 | 89.2 | **76.0** |
>
> [1]. Zhang Z, Chen W, Cheng H, et al. Divide and contrast: Source-free domain adaptation via adaptive contrastive learning[J]. Advances in Neural Information Processing Systems, 2022, 35: 5137-5149.
>
> [3]. Saito K, Kim D, Sclaroff S, et al. Semi-supervised domain adaptation via minimax entropy[C]//Proceedings of the IEEE/CVF international conference on computer vision. 2019: 8050-8058.
>
> [4]. Li J, Li G, Shi Y, et al. Cross-domain adaptive clustering for semi-supervised domain adaptation[C]//Proceedings of the IEEE/CVF Conference on Computer Vision and Pattern Recognition. 2021: 2505-2514.

---

> ### Author Response · Authors · 2023-11-22
>
> Dear reviewer. I kindly inquire whether our response has successfully addressed your uncertainties. We remain dedicated to addressing concerns you may possess with utmost eagerness. Additionally, we would be encouraged if the reviewer considers raising the score.